# Psychosocial Burden of Itch among Renal Transplant Recipients

**DOI:** 10.3390/toxins14050320

**Published:** 2022-04-30

**Authors:** Piotr K. Krajewski, Kinga Tyczyńska, Klaudia Bardowska, Piotr Olczyk, Magdalena Krajewska, Jacek C. Szepietowski

**Affiliations:** 1Department of Dermatology, Venereology and Allergology, Wroclaw Medical University, 50-368 Wroclaw, Poland; piotr.krajewski@umw.edu.pl (P.K.K.); tyczynska.king@gmail.com (K.T.); klbardowska@gmail.com (K.B.); 2Department of Nephrology and Transplantation Medicine, Wroclaw Medical University, 50-368 Wroclaw, Poland; piotr.olczyk@student.umw.edu.pl (P.O.); magdalena.krajewska@umw.edu.pl (M.K.)

**Keywords:** renal transplant, itch, chronic kidney disease, burden, depression, anxiety

## Abstract

Itch is the most common symptom of chronic dermatoses. Moreover, itch may be associated with systemic disorders. Chronic kidney disease—associated itch (CKD-aI) may affect up to 20% of renal transplant recipients (RTR). The aim of the study was to assess psychosocial burden of itch in RTR. The study was performed on a group of 129 RTR, out of which 54 (41.9%) experienced itch in the previous 3 days. A specially designed questionnaire assessing anxiety, depression, stigmatization, and quality of life was used. Results: Patients suffering from itch in the previous 3 days achieved significantly higher scores in GAD-7 (*p* < 0.001), BDI (*p* < 0.001), HADS total score (*p* < 0.001), HADS Depression (*p* = 0.004), and HADS Anxiety (*p* < 0.001). Severity of itch correlated positively with HADS, stigmatization scale, and GAD-7. Itch in RTR was associated with higher incidence of depression assessed with BDI (OR 3.7). Moreover, higher prevalence of anxiety was found among patients suffering from CKD-aI, assessed with HADS A and GAD-7 (OR 2.7 and OR 4.8, respectively) The results of our study clearly demonstrate that itch among RTR is a significant burden. Higher prevalence of depression and anxiety in this groups indicate the necessity of addressing itch relief as a part of holistic approach to patients after renal transplantation.

## 1. Introduction

Chronic kidney disease (CKD) is one of the leading global health problems affecting up to 13.4% of people in the world [1]. The most advanced stage of CKD, end stage renal disease (ESRD), is diagnosed among patients with glomerular filtration rate (GFR) lower than 15 mL/min/1.73 m^2^ [2]. It is associated with long and frequent hospitalizations, premature morbidity, and multiple complications including normocytic anemia, secondary hyperparathyroidism, mineral bone disorder, dyslipidemias, hypertension, atherosclerosis, metabolic acidosis, malignancy, and chronic itch [3,4]. The treatment of ESRD remains an important challenge for clinicians. Kidney transplantation (KTx) is a modality that offers significant survival advantage, as well as better quality of life (QoL) in comparison to maintenance hemodialysis [5,6]. Nevertheless, it is important to underline that renal transplant recipients (RTRs) may suffer from an important psychological strain due to the uncertainty about their future health and because of frequent side effects of immunosuppressive drugs [7]. It is believed that several factors, including number of comorbidities, female sex, low socioeconomic status, and level of education, may have influence on RTRs’ psychological wellbeing and prevalence of psychiatric disorders [8]. Moreover, recently we have shown that more than 20% of RTRs suffer from chronic itch [9]. This was also suggested by other authors [10,11]. To the best of our knowledge there is still no study evaluating the influence of chronic kidney disease—associated itch (CKD-aI) on patients’ mental status.

Therefore, the aim of this study was to evaluate the actual influence of itch on prevalence of depression, anxiety, and stigmatization among RTRs.

## 2. Results

The studied population consisted of 129 RTR. Out of them 54 patients (41.9%) reported suffering from itch in the last 3 days before filling out the questionnaire. There were 26 females (48.1%) and 28 males (51.9%). Patients were on average 52 ± 13.5 years old. The population was characterized as overweight with mean body mass index (BMI) of 25.9 ± 5.3 kg/m^2^. On average the subjects suffered from CKD for 19.9 ± 12.2 years and were on hemodialysis for 2.4 ± 2 years. Mean time after renal transplantation in this group was 7.8 ± 7.1 years. Only 21 (16.3) people admitted having atopic predisposition, while 22 patients (17.1%) had family history positive for atopic disorders. Although itchy patients tended to suffer from CKD for longer time, the demographic data was not statistically different from the non-itchy ones (Table 1).

Among itchy patients, the majority of them suffered from itch during hemodialysis (33 people, 61.1%). In most of the cases, the severity of itch after KTx was lower when compared to the hemodialysis period (28 people, 84.9%), while in the rest (five people, 15.2%) did not change. There was a statistically significant difference in the prevalence of itch during hemodialysis between patients who reported itch in the previous 3 days and the rest of the studied group (61.1 vs. 17.3%, respectively; *p* < 0.001).

Regarding anxiety, a statistically significant difference was found in GAD-7 score between itchy and non-itchy RTR (5.44 ± 5 points vs. 2.9 ± 4 points, respectively; *p* = 0.001). Similar findings were observed for anxiety assessed with HADS. Patients with itch scored statistically higher than rest of the studied group (5.4 ± 4.5 points vs. 2.6 ± 3.1 points; *p* < 0.001). In general, anxiety, assessed with both GAD-7 and HADS, was diagnosed in 29 and 17 itchy RTRs, respectively (53.7 and 31.4%). It was more frequent than in patients without itch (*p* < 0.001, OR = 4.8 for GAD-7 and *p* = 0.022, OR = 2.7 for HADS) (Table 2 and Table 3).

Similarly, in comparison to non-itchy RTRs, depression was significantly more common in the itchy group. Those patients scored significantly higher both in BDI (*p* < 0.001) and HADS questionnaire (*p* = 0.004). According to the cut-off points of BDI, more patients with itch met the criteria for depression (21 people, 65.6%) than in the rest of studied population (11 people, 34.4%) (OR = 3.7, *p* = 0.002). This was not observed for the HADS questionnaire (Table 2 and Table 3).

The intensity of itch according to NRS correlated positively with GAD-7 total score (r = 0.317, *p* = 0.019). Moreover, a strong positive correlation was observed for ItchyQoL total score and GAD-7 score (r = 0.561, *p* < 0.001), and 6ISS (r = 0.506, *p* = 0.001) (Figure 1).

Furthermore, the ItchyQoL total score presented a moderate, positive correlation with BDI (r = 0.293, *p* = 0.032) (Figure 2) and HADS total score (r = 0.302, *p* < 0.001).

Moderate, positive correlations were found also for 4IIQ and: GAD-7 (r = 0.320, *p* = 0.018), 6ISS (r = 0.301, *p* = 0.027), and HADS total score (r = 0.396, *p* = 0.03). Only eight patients suffering from itch (14.8%) reported any feeling of stigmatization (>1 point). Although the majority of itchy RTR (75%) were minimally stigmatized (1–2 points), patients with severe QoL impairment due to itch (2 RTR, 25%) reported high level of stigmatization (10 ± 1.41 points). Mean QoL impairment in itchy RTR group, assessed with ItchyQoL was at 41.8 ± 16.1 points. There were no statistically significant differences in psychometric assessments between males and females (detailed data not shown).

## 3. Discussion

Itch is defined as an unpleasant sensation that leads to scratching [12]. In order to become chronic, according to the International Forum to the Study of Itch (IFSI), the sensation must persist for at least 6 weeks and it needs to result in an impairment in patient’s quality of life and sleeping habits [13]. As the most common symptom in dermatology, itch is frequently associated with a variety of dermatoses [14,15,16,17,18]. Nevertheless, it may be also caused by neurological, systemic, and psychiatric disorders [19]. It has been already proven, that the presence of chronic itch (CI) is associated with an important impairment in patients’ QoL, higher prevalence of stigmatization, alexithymia, stress, mood disturbances, depression, and anxiety [11,20,21,22,23,24]. Moreover, it is frequently characterized as the most burdensome symptom of psoriasis, atopic dermatitis, and urticaria [25,26,27]. Similarly, itchy systemic disorders are commonly associated with impaired health related QoL (HRQoL) [28]. This finding is most prominent for CKD-aI [29], however, authors reported similar observations in, among others, polycythemia vera [30], HIV infection [31], hepatic disorders [32], or malignancies [33].

CKD-aI, known also as uremic itch, is a common and burdensome symptom affecting up to 35% (lifetime prevalence) of patients undergoing hemodialysis [4] and 21.3% of RTRs [9]. Its pathogenesis is yet to be fully elucidated, however it is believed to be multifactorial. Authors state that abnormal renal function, disturbances in peripheral endogenous opioid system, hyperparathyroidism, microinflammation, and neuropathy may play an important role in its development [34,35,36,37,38,39]. In view of recent publications, it is also important to underline the possible impact of the accumulation of uremic toxins, including Indoxyl Sulfate (IS) and p-Cresyl Sulfate (pCS) in the development of CKD-aI. Wang et al. [40] described a group of 112 itchy CKD patients who had significantly higher concentrations of IS and pCS in comparison to non-itchy patients (208 people). Moreover, the severity of pruritus was associated with total pCS concentration [40]. The pathomechanisms of CKD-aI in RTR are even more unclear. Its prevalence and intensity does not seem to be influenced by graft function, electrolyte disturbances, chronic inflammation, or comorbidities [9]. Moreover, it was also proven that successful kidney transplantation with good renal function may significantly lower the levels of uremic toxins often responsible for itch development [39,41]. Due to unclear pathogenesis, the treatment of CKD-aI remains an important challenge and its outcome is frequently unsatisfactory both for patients and clinicians.

Psychiatric disorders among RTRs have already been analyzed by multiple authors. However, it is important to underline that to the best of our knowledge there is no study evaluating itch and its influence on RTRs’ mental health. The prevalence of psychiatric comorbidities varies greatly amid different studies. According to metanalysis performed by Palmer et al. [42] the reported prevalence of depressive symptoms was imprecise and ranged between 2.0 and 76.5%. Nevertheless, authors stated, that the summary meta-analytical prevalence of depression could be assessed at 26.6% [42]. This is in accordance with our findings; the incidence of depression in non-itchy RTR in our study was assessed as 34.4%. However, itchy RTRs reported having symptoms of depression more frequently (65.6%), which clearly indicates the influence of itch on patients’ mental status. Similar differences were observed for the prevalence of anxiety, which has been studied by Tang et al. [43]. The authors documented that only 9% of RTR and 15% of patients on hemodialysis presented symptoms of anxiety [43]. Once again, itchy RTR in our study had significantly higher prevalence of anxiety (53.7%).

The negative influence of uremic toxins on CKD patients’ mental health and cognitive functions was described by Assem et al. in 2018 [44]. Their accumulation and major impact on both large vessels and brain microcirculation may favor the occurrence of cerebrovascular diseases, leading to cognitive disorders and dementia [44]. Although successful renal transplantation with a functioning graft would significantly lower the level of uremic toxins [41], it is still unclear if the cerebral changes are reversible [41]. Moreover, Hsu et al. [45] described a higher prevalence of depression in CKD patients with lower albumin and indoxyl sulfate (IS) levels. As authors explained, IS may have an anti-depressive effect associated with the upregulation of tryptophan hydroxylase or amino acid decarboxylase and a subsequent increase in serotonin level [45].

Interestingly, itch in RTR seems to have higher influence on mental status than in other systemic diseases e.g., diabetes mellitus Type 2 (DM2). In the study performed by Stefaniak et al. [46] authors assessed the impairment of QoL, as well as prevalence of anxiety and depression among itchy patients with DM2. Although QoL assessment in both itchy RTR and DM2 groups was similar (ItchyQoL mean score of 41.76 ± 16.05 points and 41.2 ± 13.4 points, respectively), the prevalence of depression and anxiety were significantly higher in our study. In comparison, depression, and anxiety among itchy DM2 patients were assessed as comprising 33.3 and 7.6% of patients, respectively [46], while in our study there were 65.6% itchy RTR with depression and 53.7% with anxiety. The differences might be attributed to the higher general prevalence of psychiatric comorbidities in patients after KTx. In our study 34.4% of non-itchy RTRs suffered from depression and 33.3% from anxiety, while in the study by Stefaniak et al. [46] only 1.4 and 5.7% had these, respectively. It is important to underline that RTRs may suffer from an important psychological strain due to multiple comorbidities (including DM2), polypharmacy, and uncertainty about their future health [8].

We are aware of limitations of our study. Firstly, it was a single-center, retrospective study with patients almost exclusively from one region of Poland. Moreover, the group of itchy RTRs was relatively small; however, it is important to underline that the population of RTRs is limited and the access to those patients during COVID-19 pandemic is difficult. Furthermore, we did not assess the treatment effect on patients’ mental status. Nevertheless, the experience from studies on patients who suffered from itch, among them itchy hemodialyzed patients treated with difelikefaline [47] showed that the reduction of itch intensity significantly improves patients’ quality of life. Therefore, we believe that it is possible to extrapolate the results and assume that itch relief could improve patients’ mental status. Lastly, in our study we only used itch-specific questionnaires (such as ItchyQoL, 4IIQ and 6-ISS) which do not allow to compare the results between itchy and non-itchy groups. In the future, such a study, with more general questionnaires, could enhance the limited knowledge on itch in RTR and its influence on patients’ psychosocial status

In conclusion, to the best of our knowledge, this is the first study assessing the influence of CKD-aI on RTRs mental status. The results of our study present high impairment of well-being as well as higher incidence and severity of depression and anxiety in itchy RTRs. Our study highlights the importance of itch in this population and underlines the necessity of its adequate management.

## 4. Material and Methods

### 4.1. Study Participants

The study was performed on a group of 129 renal transplant recipients treated at the Department of Nephrology and Transplantation Medicine of Wroclaw Medical University between April and December 2021. All the patients were examined by a qualified physician, and medical history and demographic data, including sex, age, body mass index (BMI), time of CKD, time on dialysis and time after renal transplantation, were collected. Moreover, each patient was asked about the atopic predisposition and atopy in the family. All the participants were asked to complete a specially designed questionnaire. Exclusion criteria included patients who were younger than 18, who were unable to cooperate or fill out the questionnaire, those with active dermatological disorders, other pruritic diseases, and non-functioning renal transplants. The study was performed in a line with guidelines for human studies, as well as the World Medical Association Declaration of Helsinki and was accepted by Ethical Committee of Wroclaw Medical University (KB-750/2021)

### 4.2. Psychosocial Burden

In order to assess psychosocial burden of CKD-aI in RTR, each patient was asked to fill out the Hospital Anxiety and Depression Scale (HADS), Generalized Anxiety Disorder—7 (GAD-7), Beck Depression Inventory (BDI) and the 6-Item Stigmatization Scale (6-ISS).

#### 4.2.1. Hospital Anxiety and Depression Scale

HADS is a 14-item self-rating questionnaire measuring symptoms of anxiety (HADS Anxiety, HADS-A) and depression (HADS Depression, HADS-D), developed by Zigmond and Snaiths in 1983 [48]. It consists of seven questions related to depression and seven questions related to anxiety. Each question is scored on a scale from 0 to 3 points. The maximum total score is 42 points, while maximum score for every subscale is 21 points. The cut off for abnormal score for whole scale is 11 points, while depression and anxiety may be diagnosed with the respective subscale scores of 8 and more. In this study, a validated Polish version of HADS was used. [49].

#### 4.2.2. Generalized Anxiety Disorder-7

GAD-7 is a commonly used tool, which screens for generalized anxiety disorder. It consists of 7 questions assessing the anxiety symptoms in the previous 2 weeks on a 4-point scale. Depending on the frequency of symptoms, patients score 0 points (not at all), 1 point (several days), 2 points (over half the days), and 3 points (nearly every day) points. Total score is calculated by adding points from each question. Cut-off points of 5, 10, and 15 points represent mild, moderate, and severe levels of anxiety [50].

#### 4.2.3. Beck Depression Inventory

BDI is a 21 multiple choice question self-reported tool for measuring the severity of depression. It was created by Beck in 1961 [51]. Each of the question is about how the patient has been feeling in the previous week. Depending on the intensity, the score ranges from 0 to 3 points. The total score is calculated by adding scores of each answer. Higher the total score more severe symptoms of depression. The cut-off scores are indicating respectively: 0–9 points—minimal depression, 10–18 points—mild depression, 19–29 points—moderate depression, 30–63 points—severe depression [51,52].

#### 4.2.4. 6-Item Stigmatization Scale

6-ISS is a dermatology-specific instrument which is used for the assessment of perceived stigmatization due to the skin disease [53]. Each patient must answer 6 questions on a scale from 0 to 3 points: “not at all”, “sometimes”, “very often” and “always”, respectively. The questionnaire identifies 6 dimensions of stigmatization including anticipation of rejection, feelings of being flawed, sensitivity to the opinions of others, guilt and shame, negative attitudes, and secretiveness. Scoring varies from 0 to 18 points, and higher the score, bigger the stigma experienced [54].

#### 4.2.5. Quality of Life

ItchyQoL is a 22-item, 3-dimensional, itch-specific quality of life (QoL) instrument assessing symptoms, functioning, and emotions created by Zeidler et al. in 2019 [55]. Each question is scored on a 5-point scale ranging from 0 points—“never” to 5 points—“all the time”. Higher total score indicates bigger impact of itch on patients’ QoL [55].

### 4.3. Itch

The severity of itch was assessed with the following instruments: numeral rating scale (NRS), 4-item itch questionnaire (4IIQ), and ItchyQoL. Worst-itch NRS (WI-NRS) was evaluated as the highest itch intensity in the previous 3 days with the use of an 11-point scale, on which 0 means no itch and 10 points means worst imaginable itch. NRS cut-off points were used as follows: 1–2 points represent mild itch, 3–6 points moderate itch, 7–8 points severe itch, and ≥9 points very severe itch [56]. 4IIQ is an instrument developed and validated by our group, which besides itch intensity (0–5 points), also assesses frequency (0–5 points) and sleep disturbances (0–6 points). The maximum score is 19 points and higher the score, higher the itch severity [57].

### 4.4. Statistical Analysis

Statistical analysis of the obtained results was performed with the use of the IBM SPSS Statistics v. 26 (SPSS INC., Chicago, IL, USA) software. All data were assessed for normal or abnormal distribution. The minimum, maximum, mean, and standard deviation were calculated. Quantitative variables were evaluated using the Mann–Whitney U test and Spearman’s or Pearson’s correlations. For qualitative data, the Chi-squared test was used.

Differences in psychosocial scoring among different itch severities were assessed with the use of the Kruskal-Wallis test and one-way analysis of variance on ranks. A two-sided p of a value lower than 5% was considered significant.

## Figures and Tables

**Figure 1 toxins-14-00320-f001:**
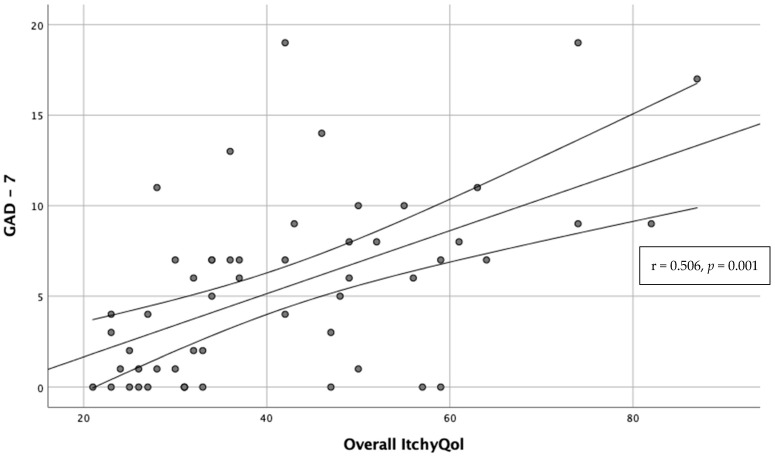
Correlation between ItchyQoL total score and GAD-7 score.

**Figure 2 toxins-14-00320-f002:**
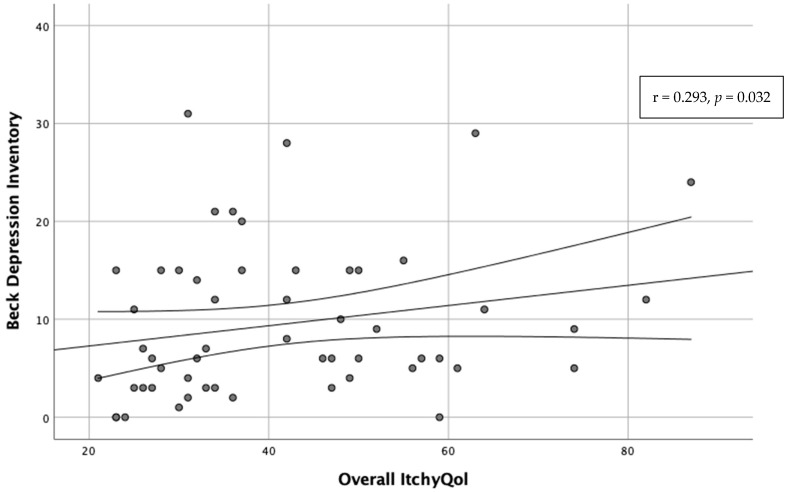
Correlation of ItchyQoL total score and BDI score.

**Table 1 toxins-14-00320-t001:** Group characteristics.

Characteristics	No Itch in Last 3 Days (*n* = 75)	Itch in Last 3 Days (*n* = 54)	*p*
Sex, *n* (%)			NA
Male	46 (61.3)	28 (51.9)	
Female	29 (38.7)	26 (48.1)	
Age (years, mean ± SD)	51.2 ± 14.6	53.1 ± 12	NS
BMI (kg/m^2^, mean ± SD)	25.7 ± 5.5	26.2 ± 5.1	NS
Time of disease (years, mean ± SD)	18.5 ± 11.5	21.8 ± 13	NS
Time on hemodialysis (years, mean ± SD)	2.1 ± 1.8	2.7 ± 2.1	NS
Time after RTx (years, mean ± SD)	7.1 ± 7.5	8.7 ± 6.5	NS
Atopy, *n* (%)	12 (16)	9 (16.7)	NS
Atopy in family, *n* (%)	11 (14.7)	11 (20.4)	NS
Itch on dialysis, *n* (%)	13 (17.3)	33 (61.1)	<0.001
Itch after RTx, *n* (%)	16 (21.3)	54 (100)	NA
New itch	9 (56.2)	21 (38.9)	
Persistent itch	7 (43.8)	33 (61.1)	
Lower severity	5 (71.4)	28 (84.9)	
Same severity	2 (28.6)	5 (15.2)	
Kidney function—eGFR (mL/min/1.73 m^2^, mean ± SD)	50.57 ± 19.44	51.08 ± 15.03	NS

*n*—number of participants; BMI—body mass index; SD—standard deviation; RTx—renal transplantation; NS—not significant; NA—not applicable; eGFR—estimated glomerular filtration rate.

**Table 2 toxins-14-00320-t002:** Differences between itchy and non-itchy RTR.

Characteristic (Mean ± SD)	No Itch in Last 3 Days (*n* = 75)	Itch in Last 3 Days (*n* = 54)	*p*
GAD-7	2.9 ± 4	5.4 ± 5	0.001
BDI	4.9 ± 5.9	9.5 ± 7.7	<0.001
HADS Total score	7.4 ± 6.12	13.2 ± 8.4	<0.001
HADS D	3 ± 2.8	4.7 ± 3.7	0.004
HADS A	2.6 ± 3.1	5.4 ± 4.5	<0.001
ItchyQoL	NA	41.8 ± 16.1	NA
6-ISS	NA	0.9 ± 2	NA

*n*—number of participants; SD—standard deviation; NA—not applicable; GAD-7—Generalized Anxiety Disorder-7; 6-ISS—6-Item Stigmatization Scale; BDI—Beck Depression Inventory; HADS—Hospital Anxiety and Depression Scale; D—depression; A—Anxiety.

**Table 3 toxins-14-00320-t003:** Frequency of depression and anxiety.

Characteristics (*n*, %)	No Itch in Last 3 Days (*n* = 75)	Itch in Last 3 Days (*n* = 54)	OR	*p*
**Anxiety**				
GAD-7	13 (32.5)	27 (67.5)	4.8	<0.001
HADS	11 (39.3)	16 (60.7)	2.7	0.022
**Depression**				
BDI	11 (34.4)	21 (65.6)	3.7	0.002
HADS	7 (41.2)	10 (58.8)	NA	NS

*n*—number of participants; NA—not applicable; NS—not significant; GAD-7—Generalized Anxiety Disorder-7; BDI—Beck Depression Inventory; HADS—Hospital Anxiety and Depression Scale; OR—odds ratio.

## Data Availability

Available on request from corresponding author.

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
