# Peer review of "Psychosocial Burden of Itch among Renal Transplant Recipients"

_toxins, 2022, doi:10.3390/toxins14050320_

Round 1
Reviewer 1 Report
The authors studied anxiety and depression extent in renal transplant recipients in relation to itch intensity.
Comments
- Lines 43-44: this sentence is not clear. It should be clarified.
- Table 1: The data presented as mean and standard deviation supposed to be normally distributed. In this case unpaired T-test (and not Mann-Whitney U test) should be applied. The authors should indicate which test has been applied.
- Lines 175-211: The numbering of these paragraphs should be corrected.
Author Response
Dear Reviewer:
Thank you very much for this positive review and for your time. Please find the answer to your question below. The changes in the manuscript were highlighted.
- Lines 43-44: this sentence is not clear. It should be clarified.
- Thank you for this comment. It was clarified
- Table 1: The data presented as mean and standard deviation supposed to be normally distributed. In this case unpaired T-test (and not Mann-Whitney U test) should be applied. The authors should indicate which test has been applied.
- Thank you for this comment. As all the data were not normally distributed, however the data was also continuous we used Mann-Whitney U test comparing means between groups.
- Lines 175-211: The numbering of these paragraphs should be corrected.
- Thank you for this remark. It was changed
Reviewer 2 Report
In this wise and clearly written study authors analyze the frequency and burden of chronic itch in a cohort of renal transplant patients. They found a frequency of about 41% with high mental and mood consequences and as a consequence a poor quality of life.
I have few comments to the authors to improve the manuscript :
-It would be useful to give values of creatinine and GFR (CKD-EPI) in the group of transplanted patients without itch and in the group with itching.
-What was the repartition of itchy patients according of CKD-stages ?
-How was the repartition of the severity of itch ?
Author Response
Dear Reviewer:
Thank you very much for the review and for your time. We have revised our manuscript according to your comments. All the introduced changes are highlighted in the manuscript.
- It would be useful to give values of creatinine and GFR (CKD-EPI) in the group of transplanted patients without itch and in the group with itching.
- Thank you very much for this comment. We did not include this in the manuscript, as the results were not different for itchy and non itchy patients. Moreover, as we mentioned in the inclusion criteria, only people with functioning graft were taken into the study. Moreover, in our previous study we have seen that the graft function (while still functioning) does not enlarge the prevalence of pruritus nor its intensity. Nevertheless, we have added eGFR function (as it is more reliable than creatinine) into the table 1.
- -What was the repartition of itchy patients according of CKD-stages ? -How was the repartition of the severity of itch ?
- Thank you for this remarks. As mentioned previously we did not find any influence of graft function on the prevalence nor the intensity of itch in RTR. Therefore, the repartition of itchy patients did not differ among different CKD stages (mostly Grade 3) and the intensity was not significantly different.
Reviewer 3 Report
Authors present an interesting study concerning the influence of pruritus on psychosocial status of KTx recipients. This is a crucial problem in solid organ recipients, and dermatological problems of transplant recipients and their role on patients well-being is not a common subject of studies, however discussed recently by Kim D and Pollock C (Epidemiology and burden of chronic kidney disease-associated pruritus. Clin Kidney J. 2021 Oct 14;14(Suppl 3):i1-i7. doi: 10.1093/ckj/sfab142). However:
1) as this is a study submitted to Toxins, I would like to see more data about mechanisms of pruritus/itch and their role in observed results;
2) You've mentioned twice (intro line 3 & discussion line 104) itch is the most common symptom of (chronic) dermatoses, can you please add some reference?
3) You did not put information in the abstract how/with what kind of questionnaires you've conducted study;
4) if you mention that 54 patients experienced itch in the previous days (lines 6-7), was it any itch? or continuos itching for 3 days?
5) please be more specific about CKD complications (lines 26-28), i.e. 'bone disorders', 'cardiovascular disesase'; in CKD we should expect 'secondary' hyperparathyroidism;
6) how many % of patients reported depression or anxiety before KTx?
7) how many % of patients had diabetes mellitus, electrolyte abnormalities (hypercalcemia, hyperphosphatemia?) what about 15.2% patients who had constant pruritus before and after KTX? have you analyzed efficacy of HD and kidney graft function? maybe patients with pruritus had impaired graft function?
8) You're repeating words in some sentences (i.e. 'neuropathy' lines 118-120);
9) please try to add more data about pruritus pathogenesis in CKD/ KTx patients;
10) do you have any data if used pruritus treatment had any (beneficial) effect on psychosocial condition of patients?
11) I think presented study is more about 'psycho-' than 'social' effect, I can't see any relationship with that in the text.
Author Response
Ad. Reviewer 3
Dear Reviewer:
Thank you very much for the review and for your time. We have revised our manuscript according to your comments. All the introduced changes are highlighted in the manuscript.
- as this is a study submitted to Toxins, I would like to see more data about mechanisms of pruritus/itch and their role in observed results;
- Thank you very much for this comment. We have added the paragraph about uremic toxins and patients’ mental health.
- You've mentioned twice (intro line 3 & discussion line 104) itch is the most common symptom of (chronic) dermatoses, can you please add some reference?
- Thank you for this comment. The references were added.
- You did not put information in the abstract how/with what kind of questionnaires you've conducted study;
- We acknowledge your remark. We did not put any kind of information about used questionnaire due to the abstract requirements of the publisher. With 200 words it is almost impossible to add all the relevant information and we wanted to focus on the results. Nevertheless, a short remark was added.
- if you mention that 54 patients experienced itch in the previous days (lines 6-7), was it any itch? or continuos itching for 3 days?
- We acknowledge your comment. It is important to underline that chronic itch is almost never continuous or generalized, therefore we could not choose patients with only continuous itch. A thorough medical history, itch history, as well as dermatological examination allow us to determine if the in the last 3 days was only acute, punctual happening or a part of a longer itch continuum. We have discarded the patients with “any itch” and included only those who, in our opinion, were suffering from CKD-aI. Moreover, as the questionnaires recall period is not very long, we tended to choose patients with recent symptoms rather than those who experienced itch a long time ago
- please be more specific about CKD complications (lines 26-28), i.e. 'bone disorders', 'cardiovascular disesase'; in CKD we should expect 'secondary' hyperparathyroidism;
- Thank you for this comment. We have clarified the complications of CKD.
- how many % of patients reported depression or anxiety before KTx?
- Thank you for this remark. We agree that this would be a very interesting observation. Nevertheless, it is only a retrospective study, we did not have the possibility to evaluate patients before and after renal transplantation, which would clearly enhance the manuscript. The only possible comparison is the one with so far published cases, which was done in the discussion section.
- how many % of patients had diabetes mellitus, electrolyte abnormalities (hypercalcemia, hyperphosphatemia?) what about 15.2% patients who had constant pruritus before and after KTX? have you analyzed efficacy of HD and kidney graft function? maybe patients with pruritus had impaired graft function?
- Thank you for this comment. Very little is known about the pathogenesis of itch in renal transplant recipients. We have published one of the first studies on itch in RTR (Krajewski et al. (2021) Front Med) and in this manuscript we tried to find a correlation between itch and well-known CKD-aI itch factors (including diabetes, hypercalcemia, hyperphosphatemia). Unfortunately none of the factors, including graft function (creatinine and eGFR) would influence the occurrence or intensity of itch. Moreover, we have added eGFR of our patients in the Table 1. There was no significant difference between itchy and non itchy group.
- You're repeating words in some sentences (i.e. 'neuropathy' lines 118-120);
- Thank you for this remark. The manuscript was once again read, and the repetitions were deleted.
- please try to add more data about pruritus pathogenesis in CKD/ KTx patients;
- We acknowledge your comment. Additional data about uremic toxins was added.
- do you have any data if used pruritus treatment had any (beneficial) effect on psychosocial condition of patients?
- We acknowledge your comment. Most of the used treatment were emollients or antihistamines prescribed by general practitioners. We did not find any difference in the prevalence nor intensity of itch in patients receiving and not receiving treatment.
- I think presented study is more about 'psycho-' than 'social' effect, I can't see any relationship with that in the text.
- Thank you for this comment. We agree that depression and anxiety represent mostly the psycho- effect. Nevertheless, it is important to remember about stigmatization, which is almost totally a social effect and quality of life, which also covers daily activity and social interactions. Therefore, we believe that the used tests cover both psycho and social effects.
Round 2
Reviewer 3 Report
Authors answered sufficiently to most of my questions, however still did not put some data about social significance of itching in this population (as in the title) or treatment effect.
Author Response
Dear Reviewer:
Thank you very much for this review and for your time. Please find the answer to your question below. The changes in the manuscript were highlighted.
- Social significance of itching in this population
- Thank you for this comment. Decrease in quality of life and stigmatization due to itch is an important problem in our group. Nevertheless, in our study we only used itch-specific questionnaire (including ItchyQoL, 4IIQ and 6-ISS) which do not allow to compare the results between itchy and non-itchy group. Probably in the future such a study with more general questionnaires could enhance the limited knowledge on itch in RTR and its influence on patients psychosocial status. We agree that it is a limitation and we have added this to our limitation section.
- Treatment effect.
- Thank you for this comment. Treatment effect was not studied in this paper. As the first study assessing the influence of itch on mental status of RTR we wanted to confirm our hypothesis that itch is important burden and have a negative effect on patients’ lives. Nevertheless, the experience from studies on patients who suffered from itch, among them itchy hemodialyzed patients treated with difelikefaline (PMID: 31702883) showed that the reduction of itch intensity significantly improves patients’ quality of life. Therefore, we believe that it is possible to extrapolate the results and assume that itch relief could improve patients’ mental status. Such phrase was added in the limitations.